# Decision-Making Styles, Prosociality, and Behavioral Difficulties in Adolescent Offenders: The Mediating Role of Life Satisfaction and Emotional Experiences

**DOI:** 10.3390/bs15010080

**Published:** 2025-01-17

**Authors:** Arcadio de Jesús Cardona-Isaza, Inmaculada Montoya-Castilla, Angela Trujillo

**Affiliations:** 1Department of Psychosocial Studies, Faculty of Psychology, Universidad del Valle, Cali 760042, Colombia; 2Department of Personality, Assessment and Psychological Treatment, Faculty of Psychology, University of Valencia, 46010 Valencia, Spain; inmaculada.montoya@uv.es; 3Faculty of Psychology and Behavioral Sciences, Universidad de La Sabana, Chía 250005, Colombia; angela.trujillo@unisabana.edu.co

**Keywords:** decision-making styles, prosocial behavior, behavioral difficulties, life satisfaction, emotional experiences, adolescent offenders

## Abstract

Research on adolescents suggests that decision-making styles, emotional experiences, and life satisfaction play a crucial role in emotional and behavioral difficulties and the development of prosocial behaviors. This study analyzed the relationship between decision-making styles, prosociality, and difficulties among adolescent offenders, as well as the mediating role of life satisfaction and emotional experiences in this relationship. A total of 457 adolescents aged from 14 to 19 years (M = 16.23; S.D. = 1.31; 32.2% female) participated in this study. The variables of interest were assessed using self-reports and descriptive, reliability, correlational, predictive, and mediation analyses were performed. A significant association was found between the study variables. Non-rational decision-making styles and negative emotional experiences influenced difficulties, whereas rational decision-making, life satisfaction, and positive emotional experiences influenced prosocial behavior. Furthermore, the results show that the relationship between rational decision-making and prosocial behavior is mediated by life satisfaction, emotional balance, and positive emotional experiences. Similarly, negative emotional experiences mediated the relationship between a hypervigilance decision-making style and emotional and behavioral difficulties. The results of this study indicate the importance of intervening in decision-making styles, emotional management, and life satisfaction in offending adolescents to decrease behavioral and emotional difficulties and favor prosocial behavior.

## 1. Introduction

### 1.1. Decision-Making, Prosocial Behavior, and Behavioral Difficulties in Adolescent Offenders

Adolescent decision-making is a complex process that is influenced by biological, cognitive, social, and emotional factors. In decision-making, biological factors are directly related to psychological factors, which are associated with maturity in adolescent development and can be negatively impacted by stressful situations and stimuli present in their context and daily life ([42]). Evidence shows that prosocial decisions involve cognitive control and social cognition neural networks, with individual differences in self-consideration versus the openness and consideration toward others affecting the degree of activation in these systems ([18]; [17]).

While adolescents can make rational decisions in deliberative contexts, they often struggle in emotionally charged situations ([2]). Recently, research on decision-making processes has shifted from solely focusing on risk perception to examining the dynamic, contextual, and emotional factors that influence adolescents’ choices ([31]). Certain adolescent characteristics, such as impulsivity, sensation-seeking, and intense emotional experiences, although potentially risky, are also considered adaptive and essential for positive development ([2]; [31]). However, risky social contexts, cognitive biases or heuristics, emotional experiences, and inadequate information processing can lead to greater exposure, vulnerability, and engagement in maladaptive behaviors ([49]; [54]; [55]; [57]; [58]). Therefore, guidance and support from parents, guardians, and educators are necessary.

The way adolescents make decisions can significantly influence their prosocial behavior and the emergence of difficulties, especially in those with a recurrent history of antisocial and delinquent behaviors ([22]; [63]). In this study, difficulties refer to emotional, behavioral, hyperactivity, and peer relationship problems that can manifest in adolescents; for example, symptoms of anxiety (emotional problems), angry outbursts or rule-breaking behaviors (conduct problems), difficulty concentrating or being excessively impulsive (hyperactivity/inattention), or experiencing social isolation and frequent conflicts with peers (relationship problems). All of these problems and difficulties can occur in adolescents and affect decision-making; however, adolescent offenders represent a group of particular interest in research due to their heightened vulnerability, the high presence of multiple mental health problems, difficulties, adversities, traumas, and the psychosocial risks they face ([47]; [51]; [67]).

Understanding the intricate mechanisms underlying their decision-making styles and their relationship with emotional well-being is crucial for designing effective interventions that promote prosocial behavior, increase well-being, improve emotional management, and reduce behavioral problems. It is considered crucial to analyze the factors that can increase emotional stability and prosocial behavior in adolescent offenders and identify those that may reduce their difficulties. This line of research is not just relevant, but it could significantly enhance the well-being of adolescent offenders, facilitate mental health interventions, and contribute to their relational processes with peers, family, and society. In this regard, we align with previous research that has shown that cognitive factors such as rational decision-making, prosocial reasoning, moral education, and emotional stability, which is a personality trait, can promote prosocial behavior and mitigate aggressive behaviors ([12]; [33]; [36]; [40]; [44]; [52]; [60]; [68]).

### 1.2. Background of the Relationship Between Decision-Making Styles, Emotional Experiences, Life Satisfaction, and Prosocial Behavior

Based on previous research and particularly on the contributions of conflict theory in decision-making ([32]; [33]), this study suggests that there is a complex interaction between decision-making styles (rational and less suitable, such as hypervigilance), life satisfaction, emotional experiences (positive, negative, and affective balance), and prosocial behavior and behavioral difficulties in adolescent offenders.

This population was chosen because adolescent offenders show greater difficulties in making rational decisions, show less prosocial behavior, show more behavioral difficulties, and show a lower life satisfaction ([9]; [28]). This study’s findings could provide a deeper understanding of these complex interactions.

Various decision-making styles have been identified, primarily from conflict theory in decision-making ([32]; [33]), and consensus among researchers suggests that rational decision-making is the most suitable style compared to less advantageous ones, such as anxious (hypervigilant) decision-making, buck-passing, or procrastination, as it is associated with better social adaptation, resilience, and mental health ([6]). Despite this common position regarding the predominance of rationality in decision-making, increasing evidence shows that emotions and the correct use of intuition play a significant role in adolescents’ prosocial behavior ([25]; [54]; [56]).

When considering which factors are more determinant in adolescent prosocial behavior—cognitive or emotional—evidence is inconclusive due to the complexity of possible interactions and the number of factors involved. However, this relationship appears to be bidirectional. For example, emotions have been reported to influence decision-making, but a specific judgment or decision depends on the interactions between cognitive mechanisms (content, depth, and objectives of thought) and motivational processes triggered by each emotion. However, the cognitive process can eventually assess the emotion, whether it is persistent or sudden, and its consequences for behavior ([38]). This indicates that the capacities for deciding on prosocial behavior reside in the decision-making process. However, emotional experiences and other factors such as empathy are crucial, especially in situations requiring immediate context decisions. Empathy in decision-making is particularly significant, as it allows individuals to understand and consider the feelings and perspectives of others ([27]; [48]; [53]; [70]). Additionally, personality traits and decision-making styles have been significantly associated with individual propensities toward emotional or cognitive information processing for decision-making ([16]).

Regarding the outcomes of decisions on behavior, one study analyzed cognitive and emotional processes as predictors of prosocial and aggressive behavior in normative adolescents and found that emotional factors were better predictors of prosocial behavior and aggression. However, it should be noted that assessing cognitive factors focuses on prosocial reasoning and the cognitive components of empathy rather than on decision-making styles ([48]). Along these lines and based on conflict theory in decision-making ([33]), another study examined the influence of decision-making styles and problem-solving ability on emotional states, life satisfaction, and subjective well-being in adolescents, finding that “the ‘vigilance style’ is a significant predictor of subjective well-being and positive affect, whereas the ‘anxious style’ is a significant predictor of negative affect” ([12]). Research shows that people display different behavioral patterns and neural responses during social decision-making according to their levels of anxiety, affecting adaptation, prosocial behavior, and social interactions ([27]; [70]).

In this sense, greater rational cognitive control in decision-making should promote social interaction and prosociality, helping to better manage daily decisions in an adaptive way. Similarly, less appropriate decision-making styles, such as hypervigilance, procrastination, and buck-passing, are likely to generate greater difficulties; however, of these, hypervigilance, with its anxious component, can have the most negative effects on emotional and behavioral problems, primarily associated with regulation difficulties ([12]).

It has been identified that there is a relationship between decision-making styles and emotional states. Positive emotions are associated with rational styles, such as vigilance, which enhance performance, accuracy, information processing, and problem-focused coping ([13]). The rational decision-making style generates greater happiness, less distress, a sense of control, and achievement; it is preferable to passive styles (e.g., procrastination and delegation), often linked to negative emotional experiences ([13]; [53]). On the other hand, hypervigilance, although associated with negative emotions like anxiety, can be useful in contexts requiring attention and thorough analysis ([59]). The above indicates that rational styles promote positive emotional experiences, passive styles (procrastination and buck-passing) tend to do the opposite, and hypervigilance may be effective in specific situations.

Studies have also suggested a positive association between positive emotional experiences and prosocial behavior ([1]; [62]). In contrast, negative experiences (e.g., childhood psychological abuse and neglect) can hinder prosocial behavior ([39]). It has been observed that the expression of prosocial behavior or negative relationships with others, manifested in emotional and behavioral difficulties, depends on the degree of balance between the functionality of the emotional experience and its expression. Positive emotions such as opportunity and affiliation (happiness, satisfaction, and hope), appreciation and self-transcendence (gratitude, awe, elevation, and compassion), and social appeal and repair (guilt, regret, and shame) promote prosocial behavior. At the same time, distress (sadness, disappointment, fear, and anxiety) and dominance and status assertion (anger, disgust, contempt, envy, and pride) can generate difficulties and negative responses and behaviors toward others ([66]). Evidence also indicates a relationship between the big five personality traits and emotional experiences. Specifically, positive emotional experiences are positively associated with extraversion, openness to experience, agreeableness, and conscientiousness, while negatively associated with neuroticism. In contrast, negative emotional experiences show the opposite associations. Furthermore, it has been observed that extraversion predicts positive emotional experiences, whereas neuroticism and lower conscientiousness predict negative emotional experiences ([34]).

Regarding life satisfaction, evidence shows that it is significantly correlated with all decision-making styles, positively with vigilance or rational styles, and negatively with buck-passing, procrastination, and hypervigilance, which represent an anxious way of making decisions ([5]; [19]). In this regard, a study with adolescents examining the relationships between different forms of victimization, prosocial experiences, and emotional well-being found that life satisfaction is related to greater prosocial behavior, emotional well-being, and positive emotional experiences, and are negatively related to negative emotional experiences ([46]).

In general, cultural studies show that positive emotional experiences are more strongly related to life satisfaction than negative emotional experiences; however, how satisfaction or emotional experiences are valued may vary according to the context ([35]). In adolescents, evidence shows a positive association between positive emotional experiences and life satisfaction, while the relationship is negative with negative emotional experiences ([61]). Furthermore, lower life satisfaction predicts an increase in maladaptive behaviors, including both externalizing and internalizing behaviors, in early adolescence ([43]).

### 1.3. The Present Study

As a background for this study, evidence shows a complex relationship between emotional experiences, decision-making, and prosocial behavior in adolescents. We highlight the importance of considering positive and negative emotional experiences and life satisfaction in understanding adolescent prosocial behavior and decision-making processes. However, the lack of prior models in the literature that clearly indicate how the relationship between decision-making styles, emotional experiences, life satisfaction, and difficulties in adolescents operates makes it challenging to establish how these relationships influence prosocial behaviors and difficulties.

To establish the hypotheses of this study, we considered conflict theory in decision-making ([32]; [33]). Decision-making for situations important to the individual involves a decisional conflict that can influence emotional experiences, potentially leading to more adaptive and prosocial behaviors. According to this theory, individuals must engage in a rational process that considers the situation to direct behavior; thus, decisional conflict implies a conscious cognitive process that evaluates the conditions related to the decision. We also drew upon previous research indicating that emotions are related to decision-making and behavior expression ([38]), and that decision-making styles, particularly the rational style, affect emotional states and life satisfaction ([12]). Furthermore, we considered evidence showing that active and rational decisions positively affect emotional states, generating greater happiness, less distress, an increased sense of control, and a sense of accomplishment ([53]) and that life satisfaction seems to contribute to prosocial behavior ([46]; [71]). For the hypothesis regarding the model explaining behavioral difficulties, we considered that less suitable decision-making styles yield poorer behavioral and adaptive outcomes ([6]) and that negative emotional states can affect prosocial behavior ([39]; [66]). Additionally, lower life satisfaction increases maladaptive behaviors ([43]), while higher satisfaction is associated with adaptive and prosocial behaviors ([46]).

Based on the above, this study aimed to analyze the mediating role of emotional experiences and life satisfaction in relation to decision-making styles, prosocial behaviors, and behavioral and emotional difficulties in adolescent offenders. It seeks to contribute to the understanding of these processes and address practical implications for interventions in decision-making and prosocial behavior in adolescents in conflict with law. As discussed, emotional experiences have an important influence on decision-making ([38]); however, we propose that the cognitive process—in this case, decision-making—should be the starting point (X) that influences behavior (Y) and facilitates interaction and regulation with emotional states. If this is the case, decision-making interventions could help adolescent offenders, particularly those in conflict with the law, in processes of emotional regulation and the prosocial orientation of their behavior.

**Hypothesis** **1.**
*Rational decision-making (X) indirectly influences prosocial behavior in adolescents (Y) through the mediation of positive emotional experiences and life satisfaction. Specifically, greater rational decision-making would be related to more positive emotional experiences (M1), which, in turn, would be related to higher life satisfaction (M2), which would ultimately be associated with higher prosocial behavior.*


**Hypothesis** **2.**
*Anxious decision-making (X) indirectly influences difficulties in adolescents (Y) through the mediation of negative emotional experiences and life satisfaction. Specifically, greater anxious decision-making would be related to more negative emotional experiences (M1), which, in turn, would be related to lower life satisfaction (M2), which would ultimately be associated with higher difficulties.*


## 2. Materials and Methods

### 2.1. Participants and Procedure

This study employed a cross-sectional design in which questionnaires were administered to 457 adolescents aged between 14 and 19 years old who were in the Juvenile Justice System in Colombia, sanctioned for various offenses. The participants in this study were adolescents residing in three socio-educational centers for antisocial behavior located in the Department of Cundinamarca and Bogotá, Colombia. A total of 510 questionnaires were distributed, and 457 valid responses were collected, resulting in an effective response rate of 89.61%. The characteristics of the participants are presented in Table 1.

The inclusion criteria were being a resident in a socio-educational center, having a confirmed legal measure for an offense, being between 14 and 19 years old, having sufficient schooling to respond to the questionnaires, and not having a psychiatric diagnosis. To ensure that the participants did not have psychiatric diagnoses, their clinical history was reviewed and confirmed with the professionals responsible for the follow-up of each case in the socio-educational centers. The exclusion criteria included not having signed consent forms, voluntary exclusion, and incomplete questionnaires.

Participants were selected by convenience sampling because of their special conditions, including confinement and state protection. The study followed the Declaration of Helsinki ([69]), Colombian Government Resolution 8430 8430 ([50]), and current data protection regulations ([14]). Consent was obtained from guardians, family advocates, and the directors of socio-educational centers, while adolescents gave written assent. Participation was voluntary, anonymous, and confidential. The questionnaires were administered in coordinated sessions of up to 30 min, with the guidance of a research team member. A total of two pilot tests with some adolescents ensured understanding of the items. Following the study, clinical psychologists and educators conducted unannounced training workshops on decision-making and recreational activities for adolescents and their families.

We calculated the sample size using the G*Power program based on multiple linear regressions (fixed model, R^2^ deviation from zero), with a significance level (alpha) of 0.01, a statistical power of 0.99, and an effect size of 0.35. The indicated sample size was *n* = 104, therefore this study met the minimum sample size requirement.

### 2.2. Measures

#### 2.2.1. Decision-Making Styles

The Melbourne Decision-Making Questionnaire (MDMQ) developed by [45] ([45]) was administered. The MDMQ assesses four decision-making styles proposed in conflict theory ([15]; [33]), which are vigilance, involving a careful, unbiased, thorough, and rational evaluation of alternatives; hypervigilance, characterized by a rushed and anxious approach to decision-making; procrastination, marked by delays in decision-making; and buck-passing, which involves leaving decisions to others and avoiding responsibility. The MDMQ consists of 22 items with three response options: “Very true for me” (score 2), “Somewhat true for me” (score 1), and “Not true for me” (score 0). It has four subscales that assess vigilance (six items, e.g., “When I make decisions, I like to gather a good amount of information”), hypervigilance (five items, e.g., “I feel as if I am under a lot of time pressure when making decisions”), buck-passing (six items, e.g., “If a decision can be made by me or someone else, I let the other person make it”), and procrastination (five items, e.g., “Even after making a decision, I delay putting it into practice”). This study used a version validated for Colombian adolescents ([10]). The Cronbach’s alpha reliability for the total scale was 0.76. We conducted a Confirmatory Factor Analysis (CFA) of the scale with the current sample, showing adequate goodness-of-fit indices (χ^2^ = 335.79, *df* = 199, *p* < 0.001), TLI = 0.89, CFI = 0.91, RMSEA = 0.039, and 90% CI (0.032, 0.046) (Comparative Fit Index—CFI, Tucker-Lewis Index—TLI, Root Mean Square Error of Approximation—RMSEA).

#### 2.2.2. Prosocial Behavior and Difficulties

The Strengths and Difficulties Questionnaire (SDQ), developed by [26] ([26]), is a brief assessment tool for measuring emotional and behavioral difficulties in children and adolescents. The SDQ consists of 25 items grouped into five subscales of five items each: emotional problems (e.g., “I am often worried”), conduct problems (e.g., “I lose my temper easily when I get angry”), hyperactivity/inattention (e.g., “I am easily distracted and have trouble concentrating”), peer problems (e.g., “I would rather be alone than with people my own age”), and prosocial behavior (e.g., “I am kind to younger children”). The first four subscales were summed to generate a total difficulty score, while the prosocial behavior subscale was interpreted separately. The Spanish version of the SDQ has been used to assess emotional and behavioral problems in Spanish and Colombian adolescents ([23]; [24]). In this study, Cronbach’s alpha was 0.74 for the total scale. The CFA of the scale with the current sample showed the following goodness-of-fit indices (χ^2^ = 734.75, *df* = 264, *p* < 0.001), TLI = 0.70, CFI = 0.73, RMSEA = 0.063, and 90% CI (0.057, 0.068).

#### 2.2.3. Life Satisfaction

The Satisfaction with Life Scale (SWLS) ([20]) is a brief Likert-type scale consisting of five items, rated on a seven-point scale ranging from “strongly disagree (1)” to “strongly agree (7)”. The total score on this scale ranges from 5 to 35, assessing an individual’s overall perceived life satisfaction, with higher scores reflecting greater satisfaction. Scores between 31 and 35 indicate that the individual is very satisfied; 26–30, satisfied; 21–25, slightly satisfied; 20, neutral; 15–19, slightly dissatisfied; 10–14, dissatisfied; and 5–9, very dissatisfied. The Spanish version validated in adolescents was used ([4]). Some items on the scale include: “So far I have gotten the important things I want in life” and “If I could live my life over, I would change almost nothing”. In this study, Cronbach’s alpha was 0.79 for this scale. The CFA of the scale with the current sample showed the following goodness-of-fit indices (χ^2^ = 19.60, *df* = 5, *p* < 0.001), TLI = 0.95, CFI = 0.97, RMSEA = 0.080; 90% CI (0.045, 0.119).

#### 2.2.4. Positive and Negative Emotional Experiences

The Scale of Positive and Negative Emotional Experiences (SPANE), developed by [21] ([21]) and validated with Colombian adolescents ([11]), is a brief tool consisting of 12 items. This scale was used to assess both positive and negative experiences over the past four weeks. The SPANE is divided into two sections: one for positive experiences (SPANE-P, e.g., “In the past 4 weeks, I had pleasant feelings”) and another for negative experiences (SPANE-N, “In the past 4 weeks, I felt feelings of sadness”), each with six items. Participants rate each item on a scale from 1 to 5, where 1 means “very rarely or never” and 5 represents “very often or always”. The results of both sections can be combined by calculating the difference between the total score of positive experiences and the total score of negative experiences, which provides the hedonic balance score (SPANE-B). In this study, Cronbach’s alpha was 0.75 for this scale. The CFA of the scale with the current sample showed the following goodness-of-fit indices (χ^2^ = 166.80, *df* = 53, *p* < 0.001), TLI = 0.92, CFI = 0.94, RMSEA = 0.069, 90% CI (0.057, 0.081).

### 2.3. Data Analysis

First, a descriptive analysis of the study variables was conducted. A Confirmatory Factor Analysis (CFA) of all the questionnaires was also performed using the maximum likelihood estimation method with Amos SPSS software (v.24) ([3]). The fit indices used for the CFA included the Comparative Fit Index (CFI) ([7]), the Tucker-Lewis Index (TLI) ([65]), the Root Mean Square Error of Approximation (RMSEA) ([64]), and 90% confidence intervals. The commonly accepted guidelines for good model fit are (a) CFI and TLI values greater than 0.90 (≥0.90 is considered a good fit), and (b) RMSEA values less than 0.08 (≤0.05 indicates a good fit, ≤0.08 is an acceptable fit, and ≤0.10 is a poor fit) ([8]). Spearman correlation analyses were conducted among all the subscales of the MDMQ, SQD, SWLS, and SPANE, along with a multiple regression to assess the predictive capacity of decision-making factors (vigilance, hypervigilance, buck-passing, and procrastination), positive and negative emotional experiences, and life satisfaction on prosocial behavior and total difficulties (emotional problems, conduct problems, hyperactivity/inattention, and relationship problems). These analyses were performed using SPSS v.26.

Subsequently, a first serial multiple mediation model (Figure 1) was conducted, with positive emotional experiences and life satisfaction as mediators between rational decision-making and prosocial behavior. We also developed a model in which affective balance was used as the first mediator; the results were similar to those obtained with positive emotional experiences (Appendix A). The second model included negative emotional experiences and life satisfaction as mediators between hypervigilance and difficulties (Figure 2). We replicated the analyses with the buck-passing and procrastination decision-making styles to observe whether these decision-making styles also showed the same effects as hypervigilance (Appendix A). For the serial multivariable mediation analyses, we used Model 6 of the PROCESS macro v.4.3 ([29]). In this type of mediation, the mediators (emotional experiences and life satisfaction) are expected to influence each other directly. The independent variable (decision-making) is assumed to affect the mediators in a serial pattern, which, in turn, influences the dependent variable (prosocial behavior and difficulties). To test the indirect effects and avoid the effects of heteroscedasticity and non-normality, a heteroscedasticity-consistent standard error estimator was used ([30]), along with a bias-corrected 95% confidence interval based on a sample of 10,000 bootstraps ([29]). Data were analyzed considering age and gender as covariates.

Three regressions are required to calculate the indirect effect when estimating parameters in serial mediation analysis. First, a regression of X on M1 is conducted to obtain parameter a1. Then, a regression of X and M1 on M2 is performed to obtain *a*_2_ and *d*_21_. In the third step, a regression of X, M1, and M2 on Y is conducted to obtain parameters *c’*, *b*_1_, and *b*_2_. Optionally, a regression of X on Y can be conducted to obtain the total effect (*c*), which can also be calculated by summing the primary indirect effects (*a*_1_*b*_1_, *a*_2_*b*_2_, *a*_1_*db*_2_) and the direct effect c. When two mediators are considered, the total effect is divided into five indirect effects: three primary and two secondaries. The primary effects include the effect of M1 (*a*_1_*b*_1_), the effect of M2 (*a*_2_*b*_2_), and the serial effect of M1 and M2 (*a*_1_*db*_2_). The secondary effects include the relationship between X and M2 (*a*_1_*d*) and M1 and Y (*db*_2_). The primary effects are grouped as the total indirect effect; if this is significant, it indicates the presence of mediation. In full mediation, the direct effect of X on Y disappears, leaving only the indirect effect through the mediator. In partial mediation, although the direct effect of X on Y decreases, it does not disappear completely, allowing both direct and indirect effects to coexist ([37]).

## 3. Results

### 3.1. Descriptive Statistics

Descriptive statistics were calculated for each of the study’s main variables. Kolmogorov–Smirnov normality tests indicated that the data do not follow a normal distribution. The descriptive results are presented in Table 2.

### 3.2. Correlation Test of Scale Scores

The main study variables were subjected to a correlation analysis, and the results are presented in Table 3. Vigilance, or rational decision-making, showed a significant positive correlation with prosocial behavior (rs = 0.31, *p* < 0.001), and hypervigilance correlated with difficulties (rs = 0.30, *p* < 0.001). Life satisfaction was positively correlated with affective balance (rs = 0.40, *p* < 0.001) and prosocial behavior (rs = 0.43, *p* < 0.001). Negative emotional experiences, in turn, were positively correlated with difficulties (rs = −0.45, *p* < 0.001) and negatively correlated with affective balance (rs = 0.69, *p* < 0.001).

### 3.3. Multiple Regressions Analyzing the Effect of Variables on Prosocial Behavior and Difficulties

Decision-making styles, life satisfaction, and emotional experiences were used as independent variables in two multiple regression models to verify whether they predicted prosocial behavior and difficulties. Age and gender were used as control variables. The model was significant in predicting prosocial behavior (R^2^ = 0.31, *F*(_9,447_) = 22.70, *p* < 0.001), where gender (female), vigilance, life satisfaction, and positive emotional experiences contributed to the explained variance. The model was significant in predicting difficulties (R^2^ = 0.32, *F*(_9,447_) = 22.93, *p* < 0.001) and gender (male), hypervigilance, buck-passing, and negative emotional experiences contributed to the explained variance (Table 4).

### 3.4. Serial Multivariable Mediation Analyses

For the serial mediation data presentation, we followed the instructions and format suggested by Hayes for studies with multiple mediators ([29]) and the unstandardized coefficients are presented.

A multiple serial mediation model was conducted using Hayes’ PROCESS macro for SPSS (Model 6, Figure 1). In the serial mediation model, where the variable rational decision-making (vigilance) is presented as a factor that affects prosocial behavior through four paths (i.e., *a*_1_*b*_1_, *a*_2_*b*_2_, *a*_1_*d*_21_*b*_2_, *c*′), with positive emotional experiences and life satisfaction as mediators. The arrows in Figure 3 show the paths of the tested model, and *a*_1_, *a*_2_, *b*_1_, *b*_2_, *d*_21_, c, and *c*′ represent the unstandardized coefficients.

The results presented in Figure 3 and Table 5 show that rational decision-making has a significant effect on positive emotional states (*a*_1_ = 0.68, 95% CI [0.50, 0.86]); in this case, age was significant (*b* = 0.58, *t*(_3,453_) = 2.94, *p* = 0.003; 95% CI [0.19, 0.97]). Additionally, rational decision-making also affected life satisfaction (*a*_2_ = 0.48, 95% CI [0.24, 0.71]). Positive emotional states, controlled by rational decision-making, had a significant impact on prosocial behavior: *b*_1_ = 0.10, 95% CI [0.06, 0.14]; similarly, life satisfaction also had an effect on prosocial behavior: *b*_2_ = 0.08; 95% CI [0.06, 0.11]. In these relationships, prosocial behavior was influenced by gender (female, *b* = 0.56, *t*(_5,451_) = 3.03, *p* = 0.002; 95% CI [0.20, 0.93]). Positive emotional experiences, controlled by rational decision-making, influenced life satisfaction (*d*_21_ = 0.49, 95% CI [0.38, 0.61]).

The total effect of rational decision-making on prosocial behavior was significant (c = R^2^ = 0.13, *F*(_3,453_) = 23.22, *p* < 0.0001, *b* = 0.25, SE = 0.36, 95% CI [0.18, 0.33]). The coefficient of the direct path between the two variables (*c’* = 0.11, 95% CI [0.04, 0.18]) continued to emerge even after adding positive emotional experiences and life satisfaction as serial mediators. In the total effect of the variables on prosocial behavior, age (*b* = 0.16, SE = 0.07, *t*(_3,453_) = 2.11, *p* = 0.04, 95% CI [0.01, 0.32]) and gender (female, *b* = 0.75, SE = 0.20, *t*(_3,453_) = 3.58, *p* = 0.0009; 95% CI [0.33, 1.15]) were significant.

The analyses confirmed the total indirect effect through the mediation of positive emotional states and life satisfaction (*a*_1_*b*_1_ + *a*_2_*b*_2_ + *a*_1_*d*_21_*b*_2_ = 0.14, SE = 0.02, 95% CI [0.10, 0.19]). Additionally, for indirect effect 1 which is rational decision-making–positive emotional experiences–prosocial behavior (*a*_1_*b*_1_ = 0.07, SE = 0.017, 95% CI [0.04, 0.11]), indirect effect 2 which is rational decision-making–life satisfaction–prosocial behavior (*a*_2_*b*_2_ = 0.041, SE = 0.013, 95% CI [0.02, 0.07]), and indirect effect 3 which is rational decision-making style and prosocial behavior through two serial mediators, positive emotional experiences and life satisfaction (*a*_1_*d*_21_*b*_2_ = 0.02, SE = 0.007, 95% CI [0.02, 0.04]). These results support Hypothesis 1. Additionally, we also used a similar model but replaced positive emotional experiences with affective balance. The results show that greater rational decision-making positively affects affective balance (M1), which, in turn, affects greater life satisfaction (M2), which would ultimately positively affect prosocial behavior. This model’s total unstandardized indirect effect was *b* = 0.12, *p* < 0.001, SE = 0.02, 95% CI [0.08, 0.16], indicating that positive emotional states and affective balance act similarly in the mediating role. The results for this model can be found in the Appendix A.

A second serial-multiple-mediation model was conducted, in which the variable anxious decision-making (hypervigilance) was presented as a factor affecting behavioral difficulties through four paths (i.e., *a*_1_*b*_1_, *a*_2_*b*_2_, *a*_1_*d*_21_*b*_2_, *c′*), with negative emotional experiences and life satisfaction as mediating variables. The arrows in Figure 4 show the paths of the tested model, *a*_1_, *a*_2_, *b*_1_, *b*_2_, *d*_21_, c, and *c′* represent the unstandardized coefficients.

The results presented in Figure 4 and Table 6 show that anxious decision-making had significant effects on negative emotional states (*a*_1_ = 0.47, 95% CI [0.25, 0.69]); in this case, gender was significant (female, *b* = 1.10, *t*(_3,453_) = 2.25, *p* = 0.024, 95% CI [0.14, 2.07]). Anxious decision-making (hypervigilance) also affected life satisfaction (*a*_2_ = 0.50, 95% CI [0.18, 0.82]). Gender and age were significant (female, *b* = 1.65, *t*(_4,452_) = 2.32, *p* = 0.020, 95% CI [0.25, 3.04]; older age, *b* = 0.81, *t*(_4,452_) = 3.04, *p* = 0.002, 95% CI [0.28, 1.34]). Negative emotional experiences, controlled by anxious decision-making, influenced lower life satisfaction (*d*_21_ = −0.18, 95% CI [−0.31, −0.04]).

Negative emotional experiences controlled by anxious decision-making had a significant impact on behavioral difficulties: *b*_1_ = 0.51, 95% CI [0.42, 0.60]. However, life satisfaction did not show any effects on behavioral difficulties: *b*_2_ = 0.02 (95% CI [−0.04, 0.08]). Behavioral difficulties related to negative emotional experiences were influenced by gender (female, *b* = 1.09, *t*(_5,451_) = 2.16, *p* = 0.030, 95% CI [0.10, 2.08]).

The total effect of anxious decision-making on behavioral difficulties was significant (c = R^2^ = 0.11, *F*(_3,453_) = 19.22; *p* < 0.0001, *b* = 0.85, SE = 0.12, 95% CI [0.61, 1.10]). The coefficient of the direct path between the two variables (*c’* = 0.60, SE = 0.11, 95% CI [0.37, 0.83]) continued to emerge even after adding negative emotional experiences and life satisfaction as serial mediators. Regarding the total effect of the variables on behavioral difficulties, gender was significant (female, *b* = 1.70, *t*(_3,453_) = 3.04, *p* = 0.002, 95% CI [0.60, 2.79]).

The analyses confirmed the total indirect effect through the mediation of negative emotional states and life satisfaction (*a*_1_*b*_1_ + *a*_2_*b*_2_ + *a*_1_*d*_21_*b*_2_ = 0.25, SE = 0.11, 95% CI [0.11, 0.41]). Additionally, the indirect effect 1 effects anxious decision-making–negative emotional experiences–behavioral difficulties (*a*_1_*b*_1_ = 0.24, SE = 0.071, 95% CI [0.11, 0.39]). Indirect effect 2 for anxious decision-making–life satisfaction–behavioral difficulties was not significant (*a*_2_*b*_2_ = 0.012, SE = 0.02, 95% CI [−0.02, 0.05]), and indirect effect 3 indicates that anxious decision-making style and behavioral difficulties through two serial mediators, negative emotional experiences and life satisfaction, were not significant (*a*_1_*d*_21_*b*_2_ = −0.0021, SE = 0.003, 95% CI [−0.01, 0.004]). These results partially support Hypothesis 2, because mediation occurs only through negative emotional experiences.

In conflict theory in decision-making ([32]; [33]), buck-passing (delegation) in decision-making and procrastination are also problematic decision-making styles. Therefore, we tested these models using both buck-passing and procrastination as antecedent variables (X). In both cases, the pattern observed in the hypervigilance model is repeated, indicating that problematic decision-making styles have a direct and positive effect on difficulties and that mediation effects occur through negative emotional states but not through life satisfaction. In these two cases, only the total indirect effect and indirect effect 1 are observed, specifically in the pathway, Ind1 equals decision-making style leading to negative emotional experiences which leads to difficulties. The unstandardized total indirect effect results for buck-passing were *b* = 0.19, *p* < 0.0001, SE = 0.06, 95% CI [0.07, 0.32], and the indirect effect 1 were *b* = 0.19, *p* < 0.0001, SE = 0.06, 95% CI [0.08, 0.32]. For procrastination, the unstandardized total indirect effect results were *b* = 0.29, *p* < 0.0001, SE = 0.070, 95% CI [0.17, 0.45], and the indirect effect 1 were *b* = 0.295, *p* < 0.0001, SE = 0.069, 95% CI [0.17, 0.44]. Full results for these models can be found in Appendix A.

The results of the two mediation models tested show that rational decision-making has a positive effect on prosocial behavior, mediated by positive emotional states and life satisfaction. In this process, both age and gender influence the level of prosocial behavior, with being female and being of an older age showing stronger effects. In contrast, anxious decision-making has a negative impact on behavioral difficulties, primarily mediated by negative emotional states.

## 4. Discussion

### 4.1. The Association Between Rational Decision-Making, Emotional Experiences, Life Satisfaction, and Prosocial Behavior

This research analyzed the relationship between decision-making styles, prosocial behavior, and behavioral difficulties in adolescent offenders, assessing whether emotional experiences (positive and negative) and life satisfaction act as serial mediators in this association. The results of the serial mediation analysis for rational decision-making and prosocial behavior reveal that the specific indirect effect of rational decision-making on prosocial behavior through emotional experiences and life satisfaction is significant and positive. In the second serial mediation model, it was observed that only negative emotional experiences mediated the relationship between hypervigilance and difficulties. In all models, the mediation is partial, with both direct and indirect effects coexisting. The results obtained help to understand how decision-making styles, emotional experiences, and life satisfaction influence prosocial behavior and difficulties in adolescent offenders. In line with previous research, it has been shown that both rational and non-rational decision-making styles are linked to different emotional and behavioral trajectories, and that emotional experiences and life satisfaction, in the case of prosocial behavior, play a significant mediating role in these relationships ([25]; [27]; [60]).

Regarding the associations, the results align with observations from previous studies (Table 3). We observed a positive relationship between rational decision-making, prosocial behavior, positive emotional experiences, and life satisfaction; these results are consistent with previous studies ([36]; [41]; [48]; [68]). As in other studies, a positive relationship was observed between rational decision-making, positive emotional experiences, and life satisfaction. It has been previously suggested that this decision-making style improves mental health and is associated with less distress, a greater sense of control, and a stronger sense of achievement ([5]; [6]; [34]; [53]).

In contrast, non-rational decision-making was positively associated with negative emotional experiences and difficulties, and negatively with life satisfaction. In the case of hypervigilance, this is explained by the fact that anxious decision-making is associated with greater adaptation difficulties, reduced prosocial behavior, impaired social interactions, higher levels of anxiety during the decision-making process, and lower life satisfaction ([19]; [27]; [70]). Interestingly, in the results of this study, anxious decision-making was related to prosocial behavior, albeit in a low but significant way, supporting the thesis that an anxious decision-making style can lead to better outcomes, as individuals may analyze the situation thoroughly to find optimal solutions, which could include prosocial actions ([59]). This finding suggests an interesting parallel: while a rational approach may facilitate empathy and altruism, anxious decision-making could inhibit prosocial behavior by generating a cycle of anxiety, distress, and fear responses. However, it could also promote the individual’s active search for solutions.

The positive relationship between hypervigilance, buck-passing, and procrastination with difficulties suggests a direct relationship with behavioral difficulties, indicating that non-rational decision-making styles could be targeted for intervention to mitigate their negative effects on difficulties. According to the regression data, hypervigilance is the decision-making style with the most significant effect on difficulties, followed by buck-passing.

In line with theoretical expectations, our results support that positive emotional experiences are positively related to life satisfaction, emotional well-being, and prosocial behavior. In contrast, negative emotional experiences are negatively related to these factors ([1]; [46]; [61]; [62]). In this regard, previous studies have observed that prosocial interactions are associated with increases in life satisfaction and positive emotional experiences ([46]), and adolescents who report more positive emotional experiences and fewer negative experiences also report greater life satisfaction ([61]) and prosocial behavior ([1]). The negative association between negative emotional experiences and prosocial behavior has also been reported in other studies ([62]), as well as the negative impact of negative emotional experiences that increase aggression responses and behavioral difficulties ([39]; [66]). Our results show that rational decision-making, positive emotional experiences, and life satisfaction positively influence prosocial behavior, with positive emotional experiences having the greatest impact (Table 4). In contrast, hypervigilance and negative emotional experiences predict emotional and behavioral difficulties in adolescent offenders, with negative emotions contributing most to their problems. We believe reducing negative emotions and improving affective balance could reduce these problems ([41]; [48]; [66]).

### 4.2. The Mediating Role of Emotional Experiences and Life Satisfaction

The results confirm Hypothesis 1 by showing that rational decision-making positively effects prosocial behavior through the mediation of positive emotional experiences and life satisfaction. The total effect (c) of rational decision-making on prosocial behavior demonstrates its overall influence, integrating both direct and indirect pathways. The direct coefficient (c’) shows that this relationship persists even when controlling for mediators, suggesting an independent component linked to cognitive processes. The observed total indirect effect highlights the mediating role of positive emotions and life satisfaction, reinforcing theories that identify these variables as key mechanisms for transforming cognitive abilities into prosocial behaviors.

Our findings indicate that rational decision-making promotes the occurrence of prosocial behaviors, which are supported by positive emotional states and life satisfaction. An important aspect to highlight is that recent research suggests that sound decision-making processes have a positive effect not only on prosocial behavior but also on mental health ([6]). The reported link between rational decision-making and improvements in mental health is especially relevant for adolescent offenders, who often face multiple mental health problems, difficulties, adversities, traumas, and psychosocial risks ([47]; [51]; [67]); this suggests that programs that teach rational decision-making skills could not only promote prosocial behaviors but also contribute to improved emotional well-being.

Regarding Hypothesis 2, the results indicate that anxious decision-making significantly predicts behavioral difficulties, but only through negative emotional experiences.

The total effect (c) of anxious decision-making on difficulties demonstrates a significant influence, with a direct effect (c’) that persists even when controlling for mediators, suggesting an independent component of hypervigilance. The total indirect effect highlights the exclusive role of negative emotions as a key mediator. At the same time, life satisfaction showed no significant impact, indicating that negative emotions primarily influence anxious decision-making on difficulties.

These results reinforce the idea that anxiety is not only a symptom but also a factor that contributes to a negative cycle of problematic behavior ([32]; [33]). Although life satisfaction did not play a significant role in mediating this relationship, the findings emphasize that negative emotions act as mediators, exacerbating the relationship between anxious decision-making and difficulties in adolescent offenders.

Evidence from previous studies indicates that people with higher levels of anxiety have more difficulty making decisions ([27]; [70]). While the evidence on this is not conclusive—since anxiety in decision-making has been observed to lead to achieving outcomes by prompting individuals to seek appropriate solutions ([59])—our results align with findings suggesting that anxious decision-making is problematic, is associated with greater difficulties, and is significantly worsened by negative emotional states ([66]). This distinction is important as it could influence how interventions are designed in this context. Similarly, our results suggest that not only is anxiety in decision-making problematic, but so are, to a lesser extent, buck-passing and procrastination (Appendix A). In this sense, intervention in offenders’ decision-making should include relevant and contextualized decision-making strategies and address these styles and practices.

### 4.3. Study Limitations and Future Directions

This study has several limitations to consider. Firstly, its cross-sectional design prevents establishing definitive causal relationships between the variables studied: the mediation effects provide clues about possible causal directions. However, longitudinal studies must confirm these relationships and determine whether decision-making styles influence emotional and behavioral changes over time. Secondly, other relevant variables of interest were not included. For example, other decision-making styles, which have gained relevance in adolescent decision-making research, were not addressed in this study, particularly the intuitive style, which plays a significant role in risk-taking and prosocial behavior ([25]; [54]; [56]). Thirdly, a limitation is self-reports, which are susceptible to biases such as social desirability. Future research could use performance tests, records, third-party reports, and observations to assess decision-making and prosocial behavior. An additional limitation is the lack of inclusion of other variables that could better explain the relationships among the variables examined in this study, such as moral reasoning, motivation, empathy, and moral disengagement, and the concrete evaluation of real conflict situations involving decision-making, emotional regulation, and behavioral responses. Including these and other aspects could provide better explanations of the phenomenon studied.

Future research should examine whether interventions designed to increase positive emotions and life satisfaction can in the long-term impact decision-making and prosocial behavior in adolescent offenders. Additionally, studies in different cultural contexts could provide a broader perspective on how differences in emotional experiences and perceptions of well-being affect decision-making styles and social behaviors ([35]). Furthermore, longitudinal and comparative studies with normative adolescents and adults could be conducted to verify whether mediation processes occur similarly or to detect variations. Similarly, it is necessary to conduct studies that include detailed analyses of gender and age differences. Although gender and age were included as control variables in this study, the significant results by gender are consistent with previous research; for example, women tend to show greater prosocial behavior. However, the differences in the sample do not allow for definitive conclusions.

### 4.4. Practical Implications

The results of this study provide a solid foundation for understanding the interrelationship between decision-making styles, emotional experiences, prosocial behavior, and difficulties in adolescent offenders. The implications are relevant for academic research and offer practical guidance for designing interventions that address behavioral problems and promote better mental health in vulnerable populations. Integrating mind-centered and emotion-focused approaches may be the most effective way to address the difficulties faced by adolescents with antisocial tendencies. Given that positive emotional experiences and life satisfaction play a crucial role in prosocial behavior, interventions focusing on emotional well-being could be key to improving behavioral outcomes in this population. Programs that promote a rational decision-making style and teach emotional regulation techniques could help to reduce emotional and behavioral difficulties and foster prosocial behaviors in adolescents with antisocial and delinquent backgrounds. Similarly, strategies aimed at reducing anxiety, buck-passing, and procrastination during the decision-making process could be effective in mitigating behavioral difficulties. Interventions with adolescents should integrate these elements to promote more adaptive emotional and behavioral outcomes in this population.

## 5. Conclusions

Despite its limitations, this study provides an understanding of the relationship between decision-making styles, emotional experiences, and life satisfaction in adolescent offenders, highlighting how these factors influence both prosocial behavior and behavioral difficulties. The results suggest that adolescent offenders who employ a rational decision-making approach tend to experience greater emotional well-being and, in turn, exhibit more prosocial behaviors. In contrast, anxious decision-making is associated with greater behavioral difficulties mediated by negative emotional experiences. This reflects that adolescents who make decisions under high anxiety, delegate decision-making, or procrastinate have more emotional and behavioral problems. These findings are consistent with previous research that highlights the influence of decision-making styles on behavioral and emotional outcomes ([1]; [6]; [27]; [46]) and suggest practical implications, such as the development of interventions focused on enhancing positive emotions and life satisfaction and promoting a rational decision-making style to reduce emotional and behavioral difficulties and promote prosocial behavior in at-risk adolescents.

## Figures and Tables

**Figure 1 behavsci-15-00080-f001:**
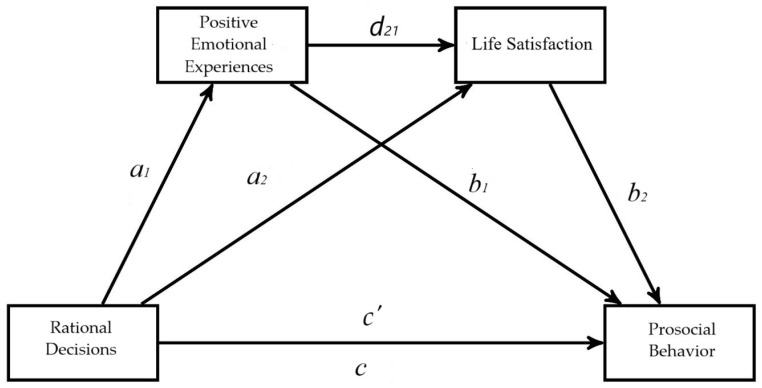
The proposed serial mediation model links rational decision-making to prosocial behavior through positive emotional experiences and life satisfaction. Note. a, b, *c*, *c′*, and d represent the path coefficients: *a*_1_ = effect of X on M1; *a*_2_ = effect of X on M2; *b*_1_ = effect of M1 on Y, controlling for the effects of X; *b*_2_ = effect of M2 on Y, by controlling for the effects of X and M1; *d*_21_ = effect of M1 on M2, controlling for X; *c* = total effect of X on Y, combining direct and indirect effects; *c*′ = represents the direct effect of X on Y, excluding any mediation through M1 or M2.

**Figure 2 behavsci-15-00080-f002:**
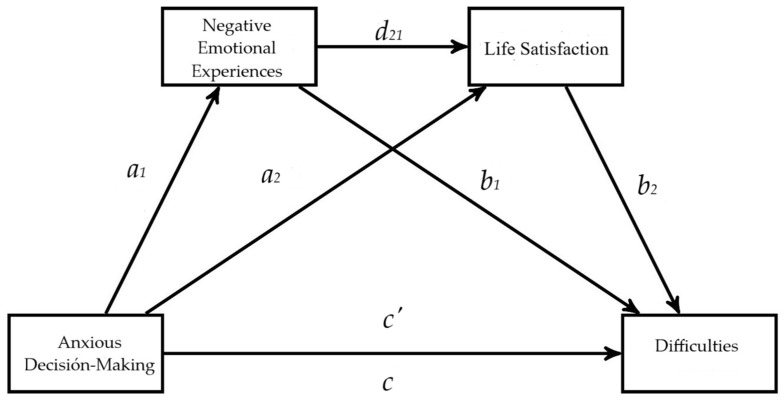
The proposed serial mediation model links anxious decision-making to behavioral difficulties in adolescents through negative emotional experiences and life satisfaction.

**Figure 3 behavsci-15-00080-f003:**
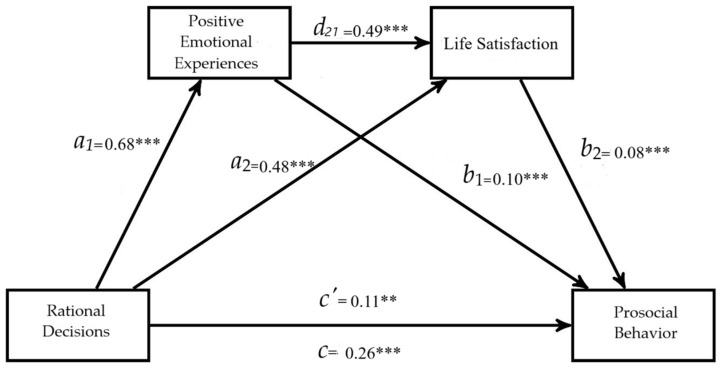
Results of the serial mediation (Model 6) with two mediators, positive emotional experiences, and life satisfaction. Coefficients are unstandardized. The path diagram with standardized coefficients is available in Appendix A. The total indirect effect was 0.14, SE = 0.022, 95% CI [0.10, 0.19]. ** *p* < 0.01, *** *p* < 0.001.

**Figure 4 behavsci-15-00080-f004:**
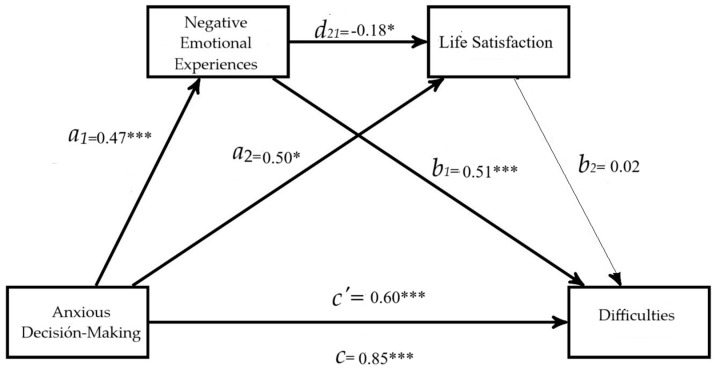
Results of the serial mediation (Model 6) with two mediators, negative emotional experiences and life satisfaction. Note. Coefficients are unstandardized. The total indirect effect 0.25, SE = 0.11, 95% CI [0.11, 0.41]. *** *p* < 0.001. * *p* < 0.05. The thin line represents regressions that are not significant. The path diagram with standardized coefficients is available in Appendix A.

**Table 1 behavsci-15-00080-t001:** Descriptive characteristics of the study participants (*n* = 457).

Characteristics	Male (%)	Female (%)	Total (%)
Age (Mean ± SD)	16.27 ± 1.24	16.16 ± 1.45	16.23 ± 1.31
Offense Type			
Theft	29.4	16.3	22.85
Drug Trafficking/Posession	45.2	59.9	52.55
Crimes Against Personal Integrity	7.4	10.9	9.15
Homicide	6.1	-	3.05
Others	11.9	12.9	12.4
Educational Level			
Completed/Primary Education	19.0	18.4	18.7
Grades 6–8 (Secondary)	61.0	42.9	51.95
Grades 9–11 (High school)	20.0	38.7	29.35
Academic Performance			
High	17.1	33.3	25.2
Medium	50.3	34.0	42.15
Low	32.6	32.7	32.65

**Table 2 behavsci-15-00080-t002:** Means, standard deviations, and the range of the variables in the study (*n* = 457).

Variables	M	SD	Range	α	ω
Gender	0.32	0.46	0–1		
Age	16.23	1.24	14–19		
Vigilance	745	2.66	0–12	0.71	0.76
Hypervigilance	4.64	2.06	0–10	0.52	0.53
Buck-passing	4.03	2.51	0–12	0.61	0.62
Procrastination	3.89	2.22	0–10	0.61	0.61
Life Satisfaction	22.01	7.21	5–35	0.79	0.79
Positive Emotional Experiences	20.68	5.58	6–30	0.86	0.86
Negative Emotional Experiences	14.81	5.03	6–30	0.81	0.81
Prosocial Behavior	5.99	2.22	0–10	0.64	0.64
Difficulties	17.08	5.89	0–40	0.73	0.74

Note. For gender, “0” = “male” and “1” = “female”; M = mean age; SD = standard deviation; range: minimum and maximum score; α = Cronbach’s alpha; and ω = McDonald’s omega.

**Table 3 behavsci-15-00080-t003:** Spearman’s correlations among the main study variables (*n* = 457).

Variables		1	2	3	4	5	6	7	8	9	10
1.	Vigilance	---									
2.	Hypervigilance	0.24 **	---								
3.	Buck-passing	−0.05	0.37 **	---							
4.	Procrastination	0.02	0.52 **	0.50 **	---						
5.	Life Satisfaction	0.33 **	0.08	−0.10 *	−0.01	---					
6.	Positive Emotional Experiences	0.33 **	0.05	−0.12 **	−0.08	0.45 **	---				
7.	Negative Emotional Experiences	−0.08	0.18 **	0.16 **	0.24 **	−0.14 **	−0.13 **	----			
8.	Affective Balance	0.29 **	−0.09	−0.20 **	−0.17 **	0.40 **	0.76 **	−0.69 **	---		
9.	Prosocial Behavior	0.30 **	0.09 *	−0.07	0.02	0.43 **	0.40 **	−0.09 *	0.35 **	---	
10.	Difficulties	−0.05	0.30 **	0.28 **	0.32 **	−0.09	−0.06	0.45 **	−0.35 **	0.13 **	---

Note. ** *p* < 0.01, * *p* < 0.05, two-tailed.

**Table 4 behavsci-15-00080-t004:** Multiple regression results for prosocial behavior and difficulties (*n* = 457).

Variables	Prosocial Behavior						Difficulties					
	B	95% CI for B		*t*	SE-B	β	B	95% CI for B		*t*	SE-B	β
		LL	UL					LL	UL			
Constant	0.18	−2.24	2.62	0.15	1.23		7.75	1.31	14.19	2.36	3.27	
Gender	0.59	0.21	0.965	3.11	0.19	0.12 **	1.24	0.25	2.23	2.47	0.503	0.09 *
Age	0.04	−0.09	0.188	0.65	0.07	0.02	−0.13	−0.50	0.23	−0.71	0.19	−0.02
Vigilance	0.10	0.03	0.178	2.89	0.03	0.13 **	−0.14	−0.33	0.04	−1.46	0.09	−0.06
Hypervigilance	0.02	−0.07	0.133	0.54	0.05	0.02	0.42	0.14	0.69	2.99	0.14	0.15 **
Buck-passing	0.03	−0.07	0.084	0.06	0.04	0.003	0.22	0.01	0.44	2.07	0.11	0.09 *
Procrastination	0.02	−0.07	0.127	0.53	0.05	0.02	0.23	−0.03	0.49	1.72	0.13	0.08
Life Satisfaction	0.08	0.05	0.113	6.13	0.01	0.28 ***	0.03	−0.003	0.11	1.05	0.03	0.04
Positive Emotional Experiences	0.10	0.06	0.139	5.69	0.02	0.26 ***	0.02	−0.07	0.12	0.45	0.04	0.02
Negative Emotional Experiences	−0.01	−0.05	0.021	−0.80	0.01	−0.03	0.47	0.38	0.57	9.88	0.04	0.41 ***

Note. B = Unstandardized coefficient; CI = confidence interval; LL = lower limit; UL = upper limit; *t* = test of variance; SE-B = standard error of B; β = standardized coefficient. * *p* < 0.05. ** *p* < 0.01. *** *p* < 0.001.

**Table 5 behavsci-15-00080-t005:** Regression coefficients, standard errors, and model summary information for positive emotional experiences and life satisfaction as mediators (*n* = 457).

					Consequent							
	M_1_ (PEE)				M_2_ (Life Satisfaction)				Y (Prosocial Behavior)			
Antecedent		Coeff.	SE	*p*		Coeff.	SE	*p*		Coeff.	SE	*p*
X (Rational decisions)	*a* _1_	0.68	0.09	<0.001	*a* _2_	0.48	0.11	<0.001	*c′*	0.11	0.03	0.001
M_1_ (PEE)		----	----	----	*d* _21_	0.49	0.05	<0.001	*b* _1_	0.10	0.018	<0.001
M_2_ (SWL)		----	----	----		----	----	----	*b* _2_	0.08	0.013	<0.001
Constant	*i* _M1_	5.90	3.32	0.071	*i* _M2_	1.08	3.95	0.783	*i_Y_*	0.19	1.16	0.864
		R^2^ = 0.13				R^2^ = 0.24				R^2^ = 0.31		
*F*(*df*)		*F*(_3,453_) = 22.90, *p* < 0.001				*F*(_4,452_) = 36.81, *p* < 0.001				*F*(_5,451_) = 40.76, *p* < 0.001		

Note. Bootstrap sample size = 10,000. Abbreviations: SE = standard error; PEE = positive emotional experiences; SWL = life satisfaction; M = mediator; *a*, *b*, *c*, *c′*, *d*, and *i* represent unstandardized regression coefficients. *F* = Fisher’s statistic; *df* = degrees of freedom; R² = Coefficient of determination.

**Table 6 behavsci-15-00080-t006:** Regression coefficients, standard errors, and model summary information for negative emotional experiences and life satisfaction as mediators (*n* = 457).

Consequent												
	M_1_ (NEE)				M_2_ (Life Satisfaction)				Y (Difficulties)			
Antecedent		Coeff.	SE	*p*		Coeff.	SE	*p*		Coeff.	SE	*p*
X (Anxious Decisions)	*a* _1_	0.47	0.11	<0.001	*a* _2_	0.50	0.16	0.002	*c′*	0.60	0.11	<0.001
M_1_ (NEE)		---	---	----	*d* _21_	−0.18	0.06	0.007	*b* _1_	0.51	0.04	<0.001
M_2_ (SWL)		---	---	---	---	---	---	---	*b* _2_	0.02	0.03	0.461
Constant	*i* _M1_	15.95	3.13	<0.001	*i* _M2_	8.57	4.62	0.064	*i_Y_*	7.51	3.28	0.022
R^2^		R^2^ = 0.05				R^2^ = 0.05				R^2^ = 0.29		
*F*(*df*)		*F*(_3,453_) = 8.57, *p* < 0.0001				*F*(_4,452_) = 6.60, *p* < 0.0001				*F*(_5,451_) = 37.40, *p* < 0.0001		

Note. Bootstrap sample size = 10,000. Abbreviations: SE = standard error; NEE = negative emotional experience; SWL = life satisfaction; M = mediator; *a*, *b*, *c*, *c′*, *d*, and *i* represent unstandardized regression coefficients. *F* = Fisher’s statistic; *df* = degrees of freedom; R² = Coefficient of determination.

## Data Availability

The data from this study can be downloaded at: https://osf.io/w8y3g/?view_only=8b202a815b18469e9a3205f596559a00.

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
