# Peer review of "Decision-Making Styles, Prosociality, and Behavioral Difficulties in Adolescent Offenders: The Mediating Role of Life Satisfaction and Emotional Experiences"

_behavsci, 2025, doi:10.3390/bs15010080_

Round 1
Reviewer 1 Report
Comments and Suggestions for Authors
11. General review of grammar and spelling is recommended. Check the correct use of terms ‘effect’ and ‘affect’ – there are some examples in which affect has been incorrectly used.
22. The introduction/background does not comment on the high prevalence of trauma and adversity for youth offending populations but it is mentioned in the discussion. Recommend just adding this into the introduction – does not need to be extensive.
33. The article could be more succinct – particularly in the discussion section which tends to repeat information about previous studies that was adequately covered in the introduction. The overview of descriptives for participants could be provided in a table, rather than paragraph (e.g., education levels etc.)
Author Response
Dear Reviewer, please review the attached file.

Reviewer 2 Report
Comments and Suggestions for Authors
Title: Decision-Making Styles, Prosociality, and Behavioral Difficulties in Adolescent Offenders: The Mediating Role of Satisfaction with Life and Emotional Experiences.
The present manuscript offers an interesting analysis of the mediating effect of positive and negative emotional experiences and satisfaction of life on the relationships between decision-making styles with prosociality and behavioral difficulties.
Authors expose rightly both the foundations and the aims of the paper as well as previous relevant literature. In this way, introduction is clearly adequate. I have just several suggestions to clarify some paragraphs:
- - I would add that emotional stability is a personality trait in line 68.
-Paragraphs about relationships between decision-making styles and positive and negative emotions (pag. 3) are somewhat confusing. It seems a list of studies but precludes finding a pattern. I will suggest rewriting them commenting all decision styles related with positive emotions altogether, and the same with negative emotions. A final sentence summarizing the main previous patterns would be also welcomed.
- - The concept of “difficulties” is largely named in some parts of the introduction. However, it would add a list of difficulties or a description to explain better what it means exactly.
In regard to method and materials:
- It should be helpful for readers to add more information about the grades. Are they equivalent to Primary, secondary, high school?
- Authors should add how was assessed psychiatric diagnosis.
- Authors should explicitly indicate what factors had the models fitted for each instrument. In any case, since all instruments are well-known, this information could be removed.
- More important for the special issue, authors must add some information about the convergent validity of positive and negative emotional experiences with personality traits (i.e., Jovanović, et al., 2020).
Results are introduced properly. I have two comments only. The heading of table 3 suggests a hierarchical (several steps) regression analysis, but this description does not fit with the description of analysis (page 8). Please, clarify. In fact, what do beta coefficients at table 3 represent? Were they beta coefficients on linear regression separately one by one, or all of them when they were entered at once? The second one is the intriguing R2 for tables 3 and 4. Note that, for instance, it is reported a coefficient of 0.683 for M1 on table 4, but it corresponds to a R2 = 0.132. Should not the R2 values in these tables be higher?
Finally, results are properly discussed, and practical implications are logically derived from findings. I have two suggestions only. I really appreciate a deep commentary about the theoretical and practical relevance of the reported differences between c and c´ coefficients. The other one is that authors argue that using an adolescent offender sample is a limitation but, in fact, it is a strong point of the paper because it guarantees variability on criteria variables as well as increase the practical impact of the paper.
Minor points:
- Nega-tively (line 152)
- I would merge the short sentences about ethical aspects on page 6 with previous or further paragraphs (lines 249-254).
- I would put “anxious decision-making” instead of inadequate to fit the previous content of the manuscript (Line 561)
- Please, remove the word important from line 653. They are usually found on this kind of research.
References:
Jovanović, V., Lazić, M., Gavrilov-Jerković, V., & Molenaar, D. (2020). The Scale of Positive and Negative Experience (SPANE): Evaluation of measurement invariance and convergent and discriminant validity. European Journal of Psychological Assessment, Vol 36(4), 2020, 694-704
Author Response

(The authors gave the same response as above.)
